# Psychological Characteristics of Women with Perinatal Depression Who Require Psychiatric Support during Pregnancy or Postpartum: A Cross-Sectional Study

**DOI:** 10.3390/ijerph20085508

**Published:** 2023-04-14

**Authors:** Grazia Terrone, Emanuela Bianciardi, Andrea Fontana, Carolina Pinci, Giulia Castellani, Irene Sferra, Anna Forastiere, Mattia Merlo, Elicio Marinucci, Fiamma Rinaldi, Marina Falanga, Daniela Pucci, Alberto Siracusano, Cinzia Niolu

**Affiliations:** 1Department of History, Cultural Heritage, Education and Society, University of Rome Tor Vergata, 00133 Rome, Italy; 2Department of Systems Medicine, University of Rome Tor Vergata, 00133 Rome, Italy; 3Department of Human Sciences, Lumsa University of Rome, 00193 Rome, Italy; 4Department of Mental Health and Pathological Addictions (DSMDP), ASL ROMA 5, 00019 Tivoli (Rome), Italy

**Keywords:** perinatal depression, body dissatisfaction, attachment style, personality traits, women health, anxiety, stressful life event

## Abstract

Antenatal depression may be distinct from postpartum depression in terms of prevalence, severity of symptoms, comorbidities, prognosis, and risk factors. Although risk factors for perinatal depression have been identified, it is unclear whether there are differences in the onset of perinatal depression (PND). This study explored the characteristics of women requiring mental health support during pregnancy or postpartum. A sample of 170 women (58% in pregnancy; 42% postpartum) who contacted the SOS-MAMMA outpatient clinic was recruited. Clinical data sheets and self-report questionnaires (EPDS, LTE-Q, BIG FIVE; ECR; BSQ; STICSA) were administered, hypothesizing possible risk factors, such as personality traits, stressful life events, body dissatisfaction, attachment style, and anxiety. Hierarchical regression models were carried out in the pregnancy (F_10;36_ = 8.075, *p* < 0.001, adjR2 = 0.877) and postpartum groups (F_10;38_ = 3.082, *p* < 0.05, adjR2 = 0.809). Recent stressful life events and conscientiousness were associated with depression in both the pregnant (29.3%, 25.5% of variance) and postpartum groups (23.8%, 20.7% of variance). In pregnant women, “openness” (11.6%), body dissatisfaction (10.2%), and anxiety (7.1%) symptoms were predictive of depression. In the postpartum group, “neuroticism” (13.8%) and insecure romantic attachment dimensions (13.4%; 9.2%) were the strongest predictors. Perinatal psychological interventions should consider the differences between mothers with depression during pregnancy and postpartum.

## 1. Introduction

Depression is one of the most frequent complications of the perinatal period and the puerperium and can occur up to one year after delivery [1]. Up to 25–35% of women experience depressive symptoms during pregnancy, while 10–20% of women develop severe and potentially fatal clinical manifestations that configure the clinical picture of perinatal depression (PND) [2].

Perinatal depression seriously compromises the health of women and offspring [3], increasing the risk of various diseases, such as preeclampsia [4] and diabetes, in pregnancy [5]. Psychiatric disorders are related to maternal mortality [6] and suicide represents 20% of the causes of death of women in the first year after childbirth [7]. Since PND affects a child’s cognitive and emotional functioning, accurate assessment and appropriate treatment are especially important.

The importance of distinguishing between different types of symptoms (e.g., depression, anxiety, and stress) and considering not only continuous but also discontinuous symptom changes from early pregnancy up to several months after childbirth have been highlighted [8]. Thus, expectant mothers have evidenced different trajectories of depression, from gradual changes to sudden and transient short-term symptoms in the first few weeks after delivery, which subside thereafter. Some women can recover quickly, while other mothers may experience an immediate increase in postpartum discomfort that lasts several months or even years after giving birth. Therefore, it is critical to consider the short- and long-term intermittent and continuous changes in depressive and anxious symptoms during the peripartum period.

Some authors have reported differences in the clinical features and prognosis of depression starting during pregnancy versus postpartum depression [9].

The timing of onset could result from different hormonal changes in pregnancy and in the postpartum period, comorbidities, such as anxiety, and other risk factors.

Thus, postpartum onset is characterized by more severe depressive symptoms, is typical of women with bipolar disorder and recurrent episodes, and is commonly associated with somatic symptoms, insomnia, and weight problems [10].

Recently, it has also been suggested to replace the DSM specification “with perinatal onset” with the distinction of times of onset: depression with pregnancy onset OR with postpartum onset [11].

The hypothesis of distinct depression occurring during pregnancy versus the postpartum period is particularly interesting and raises the question of the etiology of depression in the perinatal period [12].

Therefore, we do not pay attention to such important differences between pregnancy-onset and postpartum depression; it can be problematic both for the diagnosis and for planning the appropriate therapeutic strategies.

There are several recognized risk factors for PND, including personal and family history of psychiatric illness, single status and low social support, history of domestic abuse and violence, unplanned pregnancy, adverse life events and perceived high stress, pregnancy complications, and pregnancy loss [13].

### 1.1. Body Dissatisfaction and Perinatal Depression

Body shape and weight change rapidly during pregnancy [14]. Longitudinal studies have shown that although women may be satisfied with their body image during pregnancy, body dissatisfaction emerges after delivery [15,16]. Especially in Western countries, women are expected to monitor their weight during and after childbirth, causing significant distress to mothers [17]. In fact, prospective studies have found that body dissatisfaction was linked to depression up to 12 months postpartum [18].

### 1.2. Attachment and Perinatal Depression

The transition to motherhood has been conceptualized as a life event that activates the attachment system [19]. In the perinatal period, the mother–infant relationship is formed in which the mother participates with her attachment style [20,21]. Furthermore, insecure attachment is a significant predictor of perinatal depression [22]. While some researchers have found more significant associations of depression with different attachment styles, such as avoidant versus anxious [19], ambivalent versus avoidant [23], and preoccupied versus fearful and dismissive [21], other authors have suggested that an insecure attachment style increases the risk of depression, with no significant differences between the different subtypes [22]. An insecure attachment style is a stronger predictor of postpartum depression than prenatal depression and a history of mood disorder [22]. In a previous large study, we found that several subtypes of insecure attachment were associated with the onset of depression during pregnancy or after delivery [24]. The “need for approval” dimension, characterized by low self-confidence and excessive dependence on others, increases the risk of depression in pregnancy. On the other hand, “preoccupation with relationships”, characterized by a lack of confidence in oneself and others, was related to postpartum depression.

### 1.3. Anxiety and Perinatal Depression

Compared to perinatal depression, anxiety disorders have so far been little studied. Nonetheless, these are very common disorders, with an estimated prevalence rate of 13–20% [25]. Distinct diagnoses, anxiety, and depression are frequently comorbid, and differentiation between them can be difficult. Similar to depressive disorders, perinatal anxiety is related to the transition to motherhood. The symptoms are no different from the somatic and psychic symptoms of general anxiety, but they often focus on perinatal concerns. Anxiety can affect women’s health, that of the unborn child, the role of the mother, labor, and childbirth. Anxiety symptoms appear to decrease during mid-pregnancy before escalating again in late pregnancy and after delivery. Increased symptoms may be due to worry about impending labor, physical discomfort, and awareness that parenthood will bring life changes [26]. Especially in the third trimester of pregnancy, anxiety predicts a significant increase in depression in the postnatal period [27].

### 1.4. Stressful Life Events and Perinatal Depression

Stressful life events (SLE) are harmful or threatening events that occur in a person’s life, such as a loved one falling ill or being diagnosed with a serious illness, financial issues, unemployment, loss of something important, or being robbed. From 30 to 40% of the general population has reported at least one severe SLE in the past year [28]. SLEs are associated with the development of various somatic and mental illnesses [29,30,31]. Stressful life events have been reported to be associated with perinatal depression, with a higher risk if stressful events are numerous [32].

### 1.5. Personality Traits and Perinatal Depression

Previous studies have examined the relationship between personality traits and PND, finding that emotionally stable, extraverted, and more conscientious women are less vulnerable to depression [33].

Emotional stability indicates a tendency to experience fewer negative emotions and to be more resistant to stress [33]; therefore, emotionally stable women might be less susceptible to an escalation of psychopathological symptoms during the peripartum period. More extroverted individuals engage in social interactions more actively and experience more positive emotions [34,35,36]. Therefore, extraversion is associated with greater social support and acts as a buffer against depression. Greater conscientiousness is associated with a higher sense of mastery, purpose, meaning in life, and more effective self-regulation [33,37]. Conscientiousness has been claimed to moderate the relationship between perceived stress and depressive symptoms in the perinatal period and is known to be related to impulse and emotion control, which prevents the development of mental disorders [38].

As a result, more conscientious women may be more likely to successfully transition into motherhood. Conversely, neuroticism and interpersonal sensitivity are vulnerability factors for the development of PND [39,40,41].

However, there is a strong possibility that these distinct personality domains show significant degrees of overlap in clinical practice.

In clinical practice, only in a small number of women do the symptoms presented meet the diagnostic criteria for a major depressive episode; more frequently, the symptoms represent the epiphenomenon of a deeper discomfort, which suggests the presence of disturbances in identity integration and affective modulation. In this transition phase, affective-type symptoms arise within complex personality structures, whose areas of intrapsychic and relational functioning, which have always been problematic, become maladaptive during the delicate transition from being a daughter to being a mother.

The study of personality dimensions can ameliorate the course of perinatal depression given the possibility of guiding intervention strategies that might be focused on personality dimensions [42].

Furthermore, the predictive role of maternal personality traits with respect to the onset of depression, either before or after childbirth, is another issue to be explored. In fact, even if there is consistent literature on the effects of the different psychological dimensions we have mentioned on perinatal depression, the relationship between body dissatisfaction, stressful life events, attachment styles, and time of onset of perinatal depression has not yet been studied.

### 1.6. Aims

In this exploratory study, we compared different predictors of perinatal depression in women requiring mental health support prior (PRE-group) and post (POST-group) childbirth.

To the best of our knowledge, we can assume that there are no studies in the literature that have explored the following:The psychological features of women asking for mental support for PND before and after delivery;The role of the different predictors among women requiring mental health support during pregnancy or postpartum.

In detail, we explored whether and how body dissatisfaction, adverse life events, anxiety, the five-personality model, and attachment styles influenced perinatal depression in the two groups.

## 2. Methods

### 2.1. The Project “S.O.S. Mamma”

The “S.O.S. Mamma” Project of the University Hospital of Tor Vergata—Rome is aimed at helping women to prevent and treat psychological discomfort and psychiatric disorders of the perinatal phase.

The service can be accessed independently or upon referral from a healthcare professional. Therefore, the patients of the Mother Project “S.O.S. MAMMA” are pregnant women with current depression or with exacerbation of previous psychiatric disorders (PRE group) and women who show the onset of symptoms or their exacerbation in the postpartum period (POST group).

### 2.2. Participants and Procedure

The patients were invited to an initial meeting with a team of psychiatrists, psychologists, and residents in psychiatry. Clinical interviews and psychometric tests were administered. Appropriate pharmacological and psychological interventions were prescribed when necessary. The inclusion criteria were being over 18 years of age. The exclusion criteria were a diagnosis of psychotic spectrum disorders and poor knowledge of the language. The present research complied with APA ethical standards in treating participants, following the Ethical Principles for Medical Research Involving Human Subjects (Declaration of Helsinki). All participants provided written informed consent. The study was performed in accordance with the Helsinki declaration standards and was approved by the Institutional Ethics Review Committee of the University of Rome “Tor Vergata” on 14 March 2018.

A total of 196 women who participated in the “S.O.S. Mamma” Project were asked to participate in this study. Of these, 26 women were excluded because they had turned to S.O.S. MAMMA, as they were planning a pregnancy and were receiving drug treatment prescribed by other psychiatrists.

The final sample of 170 women was divided into two groups: 98 in pregnancy (PRE-group) and 72 in the post-partum (POST-group). Of the PRE-group, 22.2% were in the first trimester of pregnancy, 13.8% were in the second trimester, and 42.5% were in the third trimester. The mean age was 41. Most of the women were Italian (93.4%) and married (93.7%). More single women were in the POST-group. The majority of the women were employed (68.1%). Previous assisted fertilization was reported by 10.5% of women, while hormonal therapy was declared by 9.6% of the sample. Substance and alcohol abuse were relatively infrequent in the two groups, whereas tobacco use during pregnancy was present in 18.8% of the participants. A history of abortions was present in 28.3% of women.

Past psychiatric diseases affected 58.7% of women. A familiar history of psychiatric disorders was present in 54.1% of women (PRE-group 47.4%, POST-group 59.4%), and 40.6% were under pharmacological treatment (PRE-group 33.3%, POST-group 30.6%) or had been treated with psychotherapy in their lives (38.8%). Of the women, 22.4% were affected by a somatic disease, and 18.6% were on pharmacological treatment. A partner with psychiatric diseases was found in 13.5% of the PRE-group and 19.8% of the POST-group.

### 2.3. Measures

A semi-structured interview was conducted to evaluate sociodemographic data, personal, and family history of psychiatric disorders, substance and alcohol use, previous and current medical conditions, and gynecological information (previous pregnancies, abortions, assisted pregnancy, gestational week, complications of current pregnancy).

The following psychometric tests were administered.

The Edinburgh Postnatal Depression Scale (EPDS) [43] is a 10-item instrument developed as a screening tool for postpartum depression and is widely adopted for PND. Each item is scored from 0 to 3, and the total scores range from 0 to 30, with higher scores indicating higher depression. A score of 14 or higher in pregnancy and 12 or higher in postpartum has optimal sensitivity and specificity for detecting clinically relevant PND [44,45]. We used the Italian-validated version of the EPDS [45]. The EPDS scale showed internal consistency in both groups (Cronbach’s alpha 0.77 and 0.76, respectively).

The Beck Depression Inventory-II (BDI-II) [46] is a self-report questionnaire assessing symptoms of depression. The questionnaire consists of 21 items (Likert scale 0–3). It provides a total depression score and two dimensions of self-reported depression. The first dimension is the Somatic-Affective area, which concerns the manifestations of depression, such as loss of interest, loss of energy, changes in sleep and appetite, agitation, and crying. The second one is the cognitive area, which concerns the psychological symptoms of depression and the episodes of pessimism, guilt, self-criticism, and worthlessness. The scoring is interpreted as follows: 0–13 scores indicate no depressive symptoms; scores between 14–19 indicate mild depression; scores between 27–29 indicate moderate depression; and scores between 30–63 indicate severe depression. Adequate internal consistency (Cronbach’s alpha = 0.92) and evidence of convergent and discriminant validity were reported [47].

Body Shape Questionnaire (BSQ) [48]. The BSQ-34 measures body dissatisfaction. Each item is measured with a six-point Likert scale ranging from 1 “never” to 6 “always”. The scores of the items could vary from 34 to 204. Higher scores indicate greater dissatisfaction with body shape. The reliability of Cooper’s body shape questionnaire was reported as 0.88 using the test–retest method. The internal consistency of the questionnaire was confirmed by a Cronbach’s alpha value of 0.97. In the original version of the BSQ, the cut-off point for body dissatisfaction was 81.

The Experiences in Close Relationships Scale-Revised (ECR-R) [49]. The ECR-R assesses adult romantic attachment style, providing two orthogonal dimensions: a subscale of anxiety and a subscale of avoidance. High scores on the anxiety subscale indicate a tendency toward preoccupation, jealousy, and fear of abandonment, while high scores on the avoidance scale suggest uneasiness with intimacy. The questionnaire is primarily used to measure individual differences in romantic adult attachment styles and shows good psychometric properties for validity and reliability [50,51]. Several studies assessed the psychometric properties of the ECR-R, including its factorial structure, validity, and reliability in the Italian version (Cronbach’s a = 0.89 for Avoidance and 0.87 for Anxiety) [52].

The Big Five Inventory (BFI). The Italian version of the BFI [53] has proven to be reliable for both internal consistency (Cronbach’s αs ranging from 0.69 to 0.83) and temporal stability (test-retest coefficients ranging from 0.79 to 0.97). Likewise, it has proven to be valid, since scores showed the expected pattern of correlation with scores of the Big Five Questionnaire [54] convergent validity correlations ranged from 0.56 to 0.60, and discriminant validity correlations from −0.21 to 0.18. Cronbach’s αs in this study were 0.82 (Extraversion; 95% confidence interval [CI]: 0.802–0.831), 0.69 (Agreeableness; 95% CI: 0.668–0.716), 0.83 (Conscientiousness; 95% CI: 0.821–0.847), 0.80 (Neuroticism; 95% CI: 0.784–0.816), and 0.81 (Openness; 95% CI: 0.795–0.825) [55,56].

The List of Threatening Experiences (LTE-Q) [57] is a 12-item brief tool that was used to explore the presence of life adversity in the previous 12 months: e.g., severe health conditions or death of a family member, significant concerns for a beloved relative, separation or divorce, serious economic concerns, or problems with the law (one or more events vs. none). The LTE-Q was recommended for research purposes in psychiatry and social sciences. Cronbach’s alpha was 0.72.

The State-Trait Inventory for Cognitive and Somatic Anxiety (STICSA) [58] is a 21-item self-report scale discriminating between cognitive (10 items) and somatic symptoms (11 items) of both state and trait anxiety [59]. Participants responded to items on a 4-point Likert scale ranging from “almost never” to “almost always” in response to statements such as “I think others won’t approve of me” (cognitive) and “my heart beats fast” (somatic). Higher scores indicate higher levels of anxiety for both cognitive anxiety (ranging from 10 to 40) and somatic anxiety (ranging from 11 to 44). The STICSA scales are reliable in clinical and non-clinical samples (with an internal consistency of 0.90) [59,60]. It has also been considered a reliable and valid measure in older populations [61].

### 2.4. Data Analysis

SPSS software version 24 was used to compute the observed variables’ means, standard deviations, and intercorrelations. Partial correlations were examined to explore the associations between the variables. Finally, Hierarchical Regression analysis (HR) with perinatal depression scores as the dependent variable was performed, including body dissatisfaction, threatening life events and anxiety (step 1), attachment style anxiety and avoidance (step 2), and personality traits (step 3) as predictors.

Therefore, to address the issue of correlated predictors, we supplemented Hierarchical Multiple Regression with a Relative Weight Analysis approach (RWA). Specifically, RWA is a relatively new analytical strategy used to assess the importance of conceptually and empirically correlated predictors. Thus, RWA focuses on the impact of a particular predictor relative to others in the model; that is, the proportionate contribution each predictor makes to *R*^2^, considering the unique relationship with the criterion and its relationship when combined with other predictors.

## 3. Results

Descriptive statistics are reported in Table 1. There were no significant differences between the two groups in terms of age, education, employment, smoking, alcohol and substance usage, or history of psychiatric disorder.

Of the PRE-group and POST-group, 27.2% and 79.4%, respectively, reported significant depressive symptoms.

Table 2 reports the descriptive statistics of psychometric instruments.

In the PRE-group, depressive symptoms were positively correlated with Body Dissatisfaction, ECR-Avoidance subscale, ECR-Anxiety subscale, Neuroticism, Threatening Life Events, and Anxiety (Table 3). Depression negatively correlated with Extraversion, Conscientiousness, and Openness.

In the POST-group, depression positively correlates with Body Dissatisfaction, ECR-Avoidance subscale, ECR-Anxiety subscale, Neuroticism, and Threatening Life Events. Moreover, we found a negative correlation between depression and Extraversion, Conscientiousness, and Openness (Table 4).

HR analysis outlined the predictors of PRE-group depression. Three models were examined to understand which predictor explained the variance for each group. Table 5 illustrates the HR and RWA results for expectant mothers.

In the PRE-group, all three models were statistically significant. In the first model, BSQ, LTE, and STICSA were entered into the equation as predictors, explaining 18% of the total variance, F_(3; 36)_ = 15.1711; *p* < 0.001. In the second model, ECR was added to the equation, explaining 17% of the total variance, F_(5; 36)_ = 12.503; *p* < 0.001. In the third model, BSQ, LTE, STICSA, ECR, and BFI were added as predictors, explaining 87.7% of the total variance, F_(10; 36)_ = 8.075; *p* < 0.001. The RWA results evidenced the importance of Threatening Life Events in predicting prenatal maternal depression (29.3%), Body Dissatisfaction (10.2%), Anxiety (7.1%), and personality traits, such as Conscientiousness (25.5%) and Openness (11.6%), in predicting maternal depression.

In the POST-group, the first and the third models were significant. BSQ, LTE, and STICSA were entered in the first model as predictors, explaining 21% of the total variance F_(3; 38)_ = 4.287; *p* < 0.05. In the second model, ECR was added as a predictor. The second model was close to significant, F_(5; 38)_ = 3.30; *p* = 0.105. In the third model, BSQ, LTE, STICSA, ECR, and BFI were considered predictors of PND. Model 3 explained 80.9% of the total variance, F_(10; 38)_ = 3.082; *p* < 0.05. The RWA results outlined the importance of Threatening Life Events in predicting prenatal maternal depression since this predictor explained 23.8% of the variance for maternal perinatal depression. Furthermore, the RWA highlighted the importance of ECR-Attachment Avoidance (13.4%), ECR-Attachment Anxiety (9.2%), and personality traits, such as Conscientiousness (20.7%) and Neuroticism (13.8%), in predicting maternal depression (Table 6).

## 4. Discussion

In this cross-sectional study, we recruited women with perinatal depression seeking care because they were pregnant or postpartum. We hypothesized that women in need of prepartum mental health support differed from those postpartum on several psychological dimensions. Furthermore, we tested the hypothesis that different predictors of maternal depression were involved.

In line with the literature data, we found that depression was more prevalent in the postpartum period [62].

This evidence may also suggest a greater willingness of postpartum women to recognize depressive symptoms and seek support. Moreover, it is possible that depressive symptoms were not recognized during pregnancy due to guilt and fear of stigma [63].

Accordingly, it was previously reported that the proportion of recognized and treated cases of perinatal depression was lower in pregnancy than in the postnatal period [64].

Furthermore, it has been argued that postpartum depression could be considered a distinct disorder from pregnancy-onset depression due to differences in hormonal fluctuation and greater symptom severity [11].

Investigating the relative and independent weight of the predictive variables on depressive symptoms of pregnant and postpartum women, significant data emerged.

In summary, the variables most strongly associated with depression were recent stressful life events and the conscientious personality trait, which were the most significant in both the pregnant and postpartum groups.

Regarding the other variables of the study, we found significant and interesting differences in the two groups of the study.

In order of importance, we demonstrated that in pregnant women, the “openness” personality trait, body dissatisfaction, and anxiety symptoms were predictive of depression. On the other hand, in women with postpartum depression, we found that the “neuroticism” personality trait, low avoidance, and high anxiety of the romantic attachment were the strongest predictors.

Consistent with our previous findings, stressful life events in the preceding 12 months were the strongest risk factor for perinatal depression [24]. Pregnancy itself is a stressful time with many potential changes and challenges. Life’s adversities, coupled with the stress of motherhood, can facilitate the development of depression [32]. In our study, 28.3% of women miscarried, which may have contributed to their depression. In fact, it has been demonstrated that pregnancy loss has a negative impact on the mental health of the mother, regardless of whether it was a spontaneous or induced event [65].

The second strong factor associated with both pregnancy and postpartum depression was the “conscientiousness” personality trait. High conscientiousness, which characterizes responsible and reliable individuals, has been consistently and negatively correlated with depression. Conscientiousness can lead to a successful transition to motherhood. Women with high levels of conscientiousness are better able to deal with both identity and relational reorganization during the peripartum phase [66].

In the group of pregnant women, we found that the openness personality trait was negatively correlated with depression. Openness is characteristic of people who are open-minded and who deal with new situations effectively. Individuals with high openness traits are self-reliant, powerful, and tolerate being confronted with unfamiliar situations. The mothers face a new challenge: being faced with something they have never experienced before. Therefore, women with a more open mind are less prone to depression in the face of this new and unknown life experience [67].

The finding that body dissatisfaction was a predictor of depression only in pregnant women was surprising and inconsistent with the literature. Previously, it was reported that levels of body dissatisfaction were higher postpartum than in pregnancy because women were more tolerant of body changes (e.g., large abdomen, swollen feet, increased weight) perceiving their positive function [16]. However, systematic reviews highlight that women with a better body image were more likely to breastfeed [68] and that breastfeeding women had a stronger relationship with their babies [69]. In this view, body dissatisfaction was not associated with depression in postpartum women.

Finally, anxiety was a significant predictor of antenatal depression only. As mentioned, pregnancy can be a source of concern for the uncertainty of not yet having the baby in the flesh, not knowing if he will be born healthy, the fear of going through labor and delivery, and anxiety about the future role of mother [70].

In the group of postpartum women, we found that the neuroticism personality trait was correlated with depression, with a rescaled importance of 13.8%. The association between neuroticism and postpartum depression has been highlighted by numerous studies [71,72]. Twin studies have shown that depression and neuroticism share about half of the genetic variance [73]. People with high neuroticism are described as worried, insecure, hypochondriac, depressed, and suspicious. According to Han et al. [74], the neuroticism trait reduced the perception of social support and negatively affected its acceptance. Thus, a high neuroticism characterized by the propensity to experience feelings of inadequacy, low self-esteem, poor sense of autonomy, and hypersensitivity to rejection can predispose mothers to the onset or exacerbation of anxious–depressive symptoms after childbirth and to not seek help [75].

Furthermore, we found that higher insecure dimensions of romantic attachment were predictive of postpartum depression. Low levels of avoidance and high levels of anxiety in romantic relationships were associated with depression. Individuals with anxious attachment show concern about romantic relationships, fear of rejection, or being abandoned and may engage in maladaptive behavior to seek the proximity of the partner. After giving birth, mothers need the co-parent’s support to face motherhood challenges. The birth of a child determines the rapid reorganization and adaptation of the functioning of the couple. Thus, anxiety related to romantic attachment could be a risk factor for developing and maintaining feelings of hopelessness, loneliness, and lack of support, which are characteristic of depression.

Critically analyzing our results on the different predictors of depression in pregnancy and postpartum, we must underline that while before birth the predictors were mostly person centered (e.g., body dissatisfaction and anxiety), after birth, when the mother reorganized the family role with respect to the child’s presence, the relational dimension emerged as a significant predictor [24].

We point out several merits of this study. The participants required mental health help representing a specific proportion of women at high risk and with a history of psychiatric disorders. Data collection was standardized, and various psychometric instruments were performed to explore psychopathological and personality similarities and differences between the two groups.

To understand the onset and course of perinatal depression, in addition to known risk factors, such as traumatic events, it is useful to investigate the characteristics of women’s personalities and interpersonal functioning.

Such assessment would aid in intercepting women with psychological suffering and a high-risk profile but not currently symptomatic according to DSM criteria [76]. Improving the characterization of affected patients may help in establishing treatment pathways. Based on our results, it would be useful to direct the therapeutic intervention according to vulnerability factors and the perinatal period. In pregnant women, individual intervention is more appropriate, considering the burden of body dissatisfaction, anxiety, and difficulty coping with the changes of pregnancy [77]. In women with postpartum depression, it is preferable to intervene in the interpersonal sphere of the woman, strengthening the couple’s relationship and helping the mother to communicate effectively to ask for help and support in case of need [78]. In this regard, it is fundamental to take care of the mental health of partners.

Improving therapeutic interventions may also increase women’s adherence to treatment since compliance is strongly influenced by patients’ perception of benefits [79,80].

## 5. Conclusions

Peripartum depression is a clinically heterogeneous and complex disease with adverse sequelae for the mother, the child, and the partner. Our results underlined that different stages of the perinatal period were associated with specific vulnerability factors. These differences between women seeking help before and after childbirth are useful for strategically intervening in the different psychopathological dimensions even before depression manifests itself with severe symptoms. Our approach of differentiating the period of pregnancy from peripartum is useful for dealing with maternal depression in a more thoughtful and targeted way and for preventing complications.

## 6. Limitations

We recognized the limitations of our study. The results are not generalizable to the general population. Due to the cross-sectional research design, our data are a photograph of the time of assessment, and the pregnant group sample was not evaluated longitudinally, just as postpartum women were not evaluated when they were pregnant. Finally, in the regression models, we did not include demographic data that we will consider in a future and separate study to provide a more comprehensive understanding of the factors influencing perinatal depression.

## Figures and Tables

**Table 1 ijerph-20-05508-t001:** Descriptive statistics of the two groups.

		Group PRE N 98		Group POST N 72			
Variables						*t*	*p*
Mean Age (min-max; SD)		41.17 (25–44; 5.23)		40.51 (23–50; 5.59)		−0.41	0.68
		*f*	%	*f*	%	*F*	*p*
Marital status	Single	2	2	7	9.7	16.714	0.239
	Married	96	98	65	90.3		
Level of education	Primary	1	1	0	0	1.544	0.216
	Secondary	11	11	5	7		
	Diploma	47	48	43	60		
	Higher Education	39	40	24	33		
Employment status	Employed	43	44	47	65	11.894	0.458
	Not Employed	55	56	25	35		
Smoker	No	82	84	57	79	1.041	0.355
	Yes	16	16	15	21		
Alcohol use	No	96	98	69	96	2.642	0.074
	Yes	2	2	3	4		
Substance use	No	97	99	71	99	3.347	0.73
	Cannabis	0	0	1	1		
	Cocaine	0	0	0	0		
	Heroin	1	1	0	0		
Psychiatric disorder	No	56	57	39	54	1.177	0.838
	Anxiety	17	17	18	25		
	Depression	9	10	6	8		
	Depression and anxiety	11	11	9	13		
	Psychosis	3	3	0	0		
	Substance use	1	1	0	0		
	Eating disorder	1	1	0	0		

**Table 2 ijerph-20-05508-t002:** Mean values and standard deviations of the key variables and groups.

		PRE-Group N 98		POST-Group N 72	
SCALE	SUBSCALES	*M*	*SD*	*M*	*SD*
Depressive symptoms (BDI-PRE-group/EPDS-POST-group)		9.194	7.108	17.52	6.56
BSQ		69	33.42	96.18	47.3
ECR	AVOIDACE	39.44	7.33	55.71	2.7
	ANXIETY	58.33	9.92	66.63	2.4
BFI	EXTRAVERSION	51.56	8.635	48.63	8.58
	AGREEABLENESS	49.41	7.572	49.77	7.06
	CONSCIENTIOUSNESS	53.04	5.25	48.71	6.75
	NEUROTICISM	51.28	5.11	60.69	6.38
	OPENNESS	49.25	9.716	45.62	1.7
LTE		6.024	6.403	8.26	6.59
STICSA		38.86	6.19	48.868	4.4

Notes. BDI, Beck Depression Inventory; EPDS, Edinburgh Postnatal Depression Scale; BSQ, Body Shape Questionnaire; ECR, Experiences in Close Relationships; BFI, Big Five Inventory; LTE List of Threatening Experiences; STICSA, State-Inventory for Cognitive and Somatic Anxiety; SD, standard deviation.

**Table 3 ijerph-20-05508-t003:** Correlations of the key variables in the PRE-group.

	1	2	3	4	5	6	7	8	9	10	11
1 BDI	1	0.328 *	0.360 *	0.641 **	0.472 **	0.222 *	−0.333 **	−0.133	−0.292 **	0.610 **	−0.421 **
2 BSQ		1	−0.195	0.207	0.221	0.200	0.069	−0.010	−0.126	0.372 *	−0.105
3 LYTE_DIMENSIONALE			1	0.218	0.082	0.240	−0.133	−0.136	−0.248	0.209	−0.251
4 STICSA_TOT				1	0.465 **	0.419 **	−0.301	−0.172	−0.374 *	0.360 *	−0.370 *
5 ECR_AVO					1	0.270 *	−0.415 **	−0.072	−0.356 **	0.286 **	−0.469 **
6 ECR_ANX						1	−0.093	0.054	−0.186	0.427 **	−0.187
7 BIG_EXTRA							1	0.448 **	0.544 **	−0.327 **	0.533 **
8 BIG_AGREE								1	0.285 **	−0.191	0.315 **
9 BIG_CONSCIEN									1	−0.296 **	0.290 **
10 BIG_NEUROT										1	−0.326 **
11 BIG_OPENN											1

BDI Beck Depression Inventory; BSQ Body Shape Questionnaire; ECR Experiences in Close Relationships; BFI Big Five Inventory; LTE List of Threatening Experiences; STICSA State-Trait Inventory for Cognitive and Somatic Anxiety. * *p <* 0.05; ** *p <* 0.01.

**Table 4 ijerph-20-05508-t004:** Correlations of the key variables in the POST-group.

	1	2	3	4	5	6	7	8	9	10	11
1 EPDS	1	0.354 *	0.186	0.557 **	0.316 *	0.527 **	−0.426 **	−0.158	−0.366 **	0.569 **	−0.360 **
2 BSQ		1	0.178	0.461 **	0.425 **	0.433 **	0.026	−0.171	−0.241	0.506 **	0.025
3 LYTE_DIMENSIONALE			1	0.180	0.171	0.110	−0.082	−0.167	−0.017	0.080	0.216
4 STICSA_TOT				1	0.369 **	0.613 **	−0.256	0.019	−0.305 *	0.441 **	0.037
5 ECR_AVO					1	0.438 **	−0.305 *	−0.293 *	−0.441 **	0.347 **	−0.234
6 ECR_ANX						1	−0.371 **	−0.316 *	−0.377 **	0.487 **	−0.251
7 BIG_EXTRA							1	0.247	0.532 **	−0.364 **	0.504 **
8 BIG_AGREE								1	0.328 *	−0.322 *	0.332 *
9 BIG_CONSCIEN									1	−0.427 **	0.417 **
10 BIG_NEUROT										1	−0.288 *
11 BIG_OPENN											1

EPDS, Edinburgh Postnatal Depression Scale; BSQ Body Shape Questionnaire; ECR Experiences in Close Relationships; BFI Big Five Inventory; LTE List of Threatening Experiences; STICSA State-Trait Inventory for Cognitive and Somatic Anxiety. * The correlation is significant at the 0.05 level. ** The correlation is significant at the 0.01 level.

**Table 5 ijerph-20-05508-t005:** Hierarchical Regression (HR) and Relative Weight Analysis of PRE-partum group.

		ß	*t*	P	F	Raw Importance	Cumulative Raw Importance	Rescaled Importance (%)	Cumulative Rescaled Importance (%)
Dependent variable	Dependent variable: PRE−group (BDI total score POST−group (EPDS total score)					0.017	0.017	3.2	3.2
					15.1711				
Body image dissatisfaction Recent stressful life events Anxiety	BSQ	0.309	2.376	0.027		0.058	0.075	10.2	13.4
	LTE-Q	0.369	2.777	0.011		0.192	0.267	29.3	42.7
	STICSA	0.475	3.026	0.006		0.050	0.317	7.1	49.8
Adult romantic attachment style	ECR_AVOIDACE	0.063	0.319	0.753		0.019	0.336	3.2	53
	ECR_ANXIETY	−0.125	−0.882	0.388		0.019	0.355	3.4	56.4
Big Five subscales (Personality traits)	BFI_EXTRAVERSION	−0.208	−0.951	0.353		0.003	0.358	1.3	57.7
	BFI_AGREEABLENESS	−0.040	−0.307	0.762		0.026	0.384	2.1	59.8
	BFI_CONSCIENTIOUSNESS	−0.052	−0.381	0.041		0.159	0.543	25.5	85.3
	BFI_NEUROTICISM	0.271	1.979	0.411		0.037	0.68	3.1	88.4
	BFI_OPENNESS	−0.018	−0.095	0.045		0.197	0.877	11.6	100
	*R*2					0.877		100	

Notes. Rescaled importance (%) was computed by dividing the relative weights by the total *R*2 and multiplying by 100. BDI: Beck Depression Inventory; EPDS: Edinburgh Postnatal Depression Scale; BSQ Body Shape Questionnaire; ECR: Experiences in Close Relationships; BFI: Big Five Inventory; LTE-Q: List of Threatening Experiences Questionnaire; STICSA: State-Trait Inventory for Cognitive and Somatic Anxiety.

**Table 6 ijerph-20-05508-t006:** Hierarchical Regression (HR) and Relative Weight Analysis in the POST-partum group.

		ß	*t*	P	F	Raw Importance	Cumulative Raw Importance	Rescaled Importance (%)	Cumulative Resacled Importance (%)
Dependent variable	Dependent variable: PRE-group (BDI total score POST-group (EPDS total score)					0.029	0.029	4.3	4.3
					3.082				
Body image dissatisfaction Recent stressful life events Anxiety	BSQ	0.067	0.320	0.752		0.018	0.047	3	7.3
	LTE-Q	0.137	0.771	0.044		0.130	0.177	23.8	31.1
	STICSA	0.427	1.930	0.066		0.053	0.23	2.5	33.6
Adult romantic attachment style	ECR_AVOIDACE	−0.164	−0.789	0.043		0.061	0.291	13.4	47
	ECR_ANXIETY	0.247	0.981	0.033		0.052	0.343	9.2	56.2
Big Five subscales (Personality traits)	BFI_EXTRAVERSION	−0.104	−0.446	0.660		0.017	0.35	2.1	58.3
	BFI_AGREEABLENESS	−0.037	−0.179	0.860		0.019	0.369	3.1	61.4
	BFI_CONSCIENTIOUSNESS	−0.102	−0.552	0.047		0.116	0.485	20.7	82.1
	BFI_NEUROTICISM	0.379	1.716	0.010		0.177	0.662	13.8	95.9
	BFI_OPENNESS	−0.222	−0.989	0.333		0.147	0.809	4.1	100
	*R*2					0.809		100	

Notes. Rescaled importance (%) was computed by dividing the relative weights by the total *R*2 and multiplying by 100. BDI: Beck Depression Inventory; EPDS: Edinburgh Postnatal Depression Scale; BSQ Body Shape Questionnaire; ECR: Experiences in Close Relationships; BFI: Big Five Inventory; LTE-Q: List of Threatening Experiences Questionnaire; STICSA: State-Trait Inventory for Cognitive and Somatic Anxiety.

## Data Availability

The dataset is available from the corresponding author upon reasonable request.

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
