# Peer review of "Psychological Characteristics of Women with Perinatal Depression Who Require Psychiatric Support during Pregnancy or Postpartum: A Cross-Sectional Study"

_ijerph, 2023, doi:10.3390/ijerph20085508_

Round 1

Reviewer 1 Report

Comments 1.

Abstact. the authors should mention that a hierarchical regression model was carried out and the use of the instrument "The big five inventory" which has a higher weight in the regression. I understand that the journal limits the abstract to 200 words, however my suggestion is that the abstract could be restructured and omit some paragraphs, eg lines 14-17, start with “This study explored…”.

Comments 2. Methods.

2.1. The total sample of the study was 170 women; however, it is not mentioned what is the actual/total number of women who are cared for in the facility, what percentage these 170 women represent of the total number of women cared for. Based on the above, it should be mentioned why some women decided not to participate, and if this is the case, mention this in the limitations of the discussion.

2.2. Ethics. On lines 152-154, add the name of the ethics committee or committees that approved the study, as well as the identifier, or code, of the document where the research was approved.

2.3. Line 161: is it PD or PND?

Comments 3. Results.

3.1. It is not clear whether other variables such as age, education, employment, marital status, etc. were included in the hierarchical regression models. In methodology, lines 218-221, it is not mentioned, and this omission indicates that they were not included. If so, why do you think that these variables have no influence on the dependent variables BDI and EPDS?

3.2. In Table 1, when you are performing a Chi-square test to compare the PRE and POST groups, it is not correct to use this test in some cases since it is an asymptotic test that requires the expected values of the cells to be greater than or equal to 5 which is not true in some cases, you should use Fisher's exact test. In addition, some variables are missing from this table that are mentioned in lines 231-232, 235, 236-243, and more importantly, lines 246-247 that mention the prevalence of depression (27.2 and 79.4%) that do not appear anywhere, as well as the history of abortions (lines 235) that is also mentioned in lines 328 and 329 also does not appear in any table.

3.3. In Table 2, although it could be methodologically wrong to make a comparison of means between the PRE and POST groups of depressive symptoms given that the BDI and EPDS scales are used, it could be interesting to see the differences in the rest of the scales as BSQ, ECR, BFI, LTE, and STICSA, for example, a BSQ t-test between the two groups reveals a significant difference of -27.18 (p=0.0001).

3.4. In tables 3a and 3b, according to the hierarchies of the regression model showed in table 4, change columns 5,6,7,8 and 9 of the BFI block and columns 10 and 11 of LTE and STICSA, so that the order be 1. BSQ, 2. LTE, 3. STICSA, 4. ECR_AVOIDANCE, 5. ECR_ANXIETY, 6. BFI_EXTRAVERSION, 7. BFI_AGREEABLENESS. 8. BFI_CONSCIENTIOUSNESS, 9. BFI_NEUTICISM, 10. BFI_OPENESS, according to table 4a and 4b.

3.5. In tables 4 and 5, remove the t value column, and add between columns 6 and 7 (raw importance, rescaled importance) the F-Fisher values of each block. This is more informative for understanding lines 279-283 of box 4a and 288-292 of box 4b. And if possible, also the cumulative sum of the R's in each block: 0.058, 0.250 (.058+.192), 0.300, etc.

3.6. Homogenize the decimals in all the boxes, sometimes it is put “,” or “.” to separate the decimals, and can be confusing to read.

3.7. In Table 4a, “POST-group (EPDS total score)” could be omitted as it is confusing at first glance, as a suggestion, the title could be changed to “PRE-partum group: Hierarchical Regression (HR) and Relative Weight Analysis on BDI total score”. Something similar can be done with frame 4b.

Author Response

Comments 1.

Abstact. the authors should mention that a hierarchical regression model was carried out and the use of the instrument "The big five inventory" which has a higher weight in the regression. I understand that the journal limits the abstract to 200 words, however my suggestion is that the abstract could be restructured and omit some paragraphs, eg lines 14-17, start with “This study explored…”.

We thank the reviewer for the suggestion; accordingly, we wrote a new abstract including the main results

Comments 2. Methods. 

2.1. The total sample of the study was 170 women; however, it is not mentioned what is the actual/total number of women who are cared for in the facility, what percentage these 170 women represent of the total number of women cared for. Based on the above, it should be mentioned why some women decided not to participate, and if this is the case, mention this in the limitations of the discussion.

We thank the reviewer. Following the suggestion, in this revised version we mentioned the percentage of patients approached and the final sample.

2.2. Ethics. On lines 152-154, add the name of the ethics committee or committees that approved the study, as well as the identifier, or code, of the document where the research was approved.

We thank the reviewer for the suggestion; accordingly, we insert data about the Ethic committee approval.

2.3. Line 161: is it PD or PND?

We are sorry for the typo that we corrected.

Comments 3. Results.

3.1. It is not clear whether other variables such as age, education, employment, marital status, etc. were included in the hierarchical regression models. In methodology, lines 218-221, it is not mentioned, and this omission indicates that they were not included. If so, why do you think that these variables have no influence on the dependent variables BDI and EPDS?

Thank you for raising this important point. We do not insert in the regression models these variables because we concentrate our exploration in assessing how body dissatisfaction, life threaten events, attachment, and personality traits differently predict the perinatal depression in the PRE-group and in the POST-group. We agree with the reviewer that age, education, etc… could play an important role in contributing to perinatal depression, but we do not consider them in the model because we focused only on body dissatisfaction, life threaten events, attachment and personality. Furthermore, we evidenced that no statistical differences occurred between PRE-group and POST-group on these demographic variables. However, we acknowledge the importance of considering demographic variables, such as age, education, employment, and marital status, as potential confounding factors or covariates in our analysis. These variables may have an indirect effect on the relationship between our primary predictors and the dependent variables. By excluding these variables, we sought to provide a more focused investigation of our primary research questions, with the understanding that future research could further explore the potential impact of these additional factors. In light of your feedback, we address the limitation of not including these variables in our current study in the “6. Limitation” section of our manuscript. We will emphasize the need for future research to incorporate these variables into the regression models to further elucidate the relationship between our predictors of interest and the dependent variables.

3.2. In Table 1, when you are performing a Chi-square test to compare the PRE and POST groups, it is not correct to use this test in some cases since it is an asymptotic test that requires the expected values of the cells to be greater than or equal to 5 which is not true in some cases, you should use Fisher's exact test. In addition, some variables are missing from this table that are mentioned in lines 231-232, 235, 236-243, and more importantly, lines 246-247 that mention the prevalence of depression (27.2 and 79.4%) that do not appear anywhere, as well as the history of abortions (lines 235) that is also mentioned in lines 328 and 329 also does not appear in any table.

We are grateful to Reviewer 1 to noticing this point. We adjusted this mistake using Fisher’s exact test when needed.

Considering your observation, the variables are missing from this table that are mentioned in lines 231-232, 235, 236-243 have been moved to the paragraph “Participants….” because they were used to describe the clinical aspects of the sample recruited for the present study. We posit that a comprehensive and rigorous investigation should be allocated to these variables, which we intend to conduct in subsequent research endeavors.

3.3. In Table 2, although it could be methodologically wrong to make a comparison of means between the PRE and POST groups of depressive symptoms given that the BDI and EPDS scales are used, it could be interesting to see the differences in the rest of the scales as BSQ, ECR, BFI, LTE, and STICSA, for example, a BSQ t-test between the two groups reveals a significant difference of -27.18 (p=0.0001).

We thank the Reviewer 1 for raising this intriguing and important point. We agree that is methodologically wrong comparing BDI and EPDS, that’s why we did not compare this two scales. By the way, even if interesting, we preferred to not test the statistical differences between the two groups on BSQ, ECR, etc.. because the aims of our study is exploring the different weights of diverse predictors on perinatal depression in these two groups rather than comparing the main differences between the two groups. Thus, we prefer to not report the t-test and p-values on Table 2, because it could be misleading and confounding for the reader that could think, as example, that the group PRE and the group POST were paired groups or the same group assessed on two time points (pre-post).

3.4. In tables 3a and 3b, according to the hierarchies of the regression model showed in table 4, change columns 5,6,7,8 and 9 of the BFI block and columns 10 and 11 of LTE and STICSA, so that the order be 1. BSQ, 2. LTE, 3. STICSA, 4. ECR_AVOIDANCE, 5. ECR_ANXIETY, 6. BFI_EXTRAVERSION, 7. BFI_AGREEABLENESS. 8. BFI_CONSCIENTIOUSNESS, 9. BFI_NEUTICISM, 10. BFI_OPENESS, according to table 4a and 4b.

We modified the Tables as indicated

3.5. In tables 4 and 5, remove the t value column, and add between columns 6 and 7 (raw importance, rescaled importance) the F-Fisher values of each block. This is more informative for understanding lines 279-283 of box 4a and 288-292 of box 4b. And if possible, also the cumulative sum of the R's in each block: 0.058, 0.250 (.058+.192), 0.300, etc.

We modified the Tables as indicated

3.6. Homogenize the decimals in all the boxes, sometimes it is put “,” or “.” to separate the decimals, and can be confusing to read.

Thank you, we modified this point.

3.7. In Table 4a, “POST-group (EPDS total score)” could be omitted as it is confusing at first glance, as a suggestion, the title could be changed to “PRE-partum group: Hierarchical Regression (HR) and Relative Weight Analysis on BDI total score”. Something similar can be done with frame 4b.

Thank you, we modified the Title.

Reviewer 2 Report

The article covers the scientifically and practically important topic of perinatal depression. I consider the question of the determinants of perinatal depression to be essential and in need of both research and scientific reflection. The text submitted, although based on important research, needs improvement both on the surface side (English language, punctuation, standardization of the recording of statistics, the decision as to whether to write "0.2" or ".2", Cronbach's a vs. alpha, etc.), and on the structure of the scientific statement and the very issue of depression shown here. I recommend proof-reading, and I elaborate on the latter two issues in this review: 

Abstract:

- provide a rationale for why the two groups will be compared (why they should differ in the factors determining the onset of depression);

- expand on the vague phrase "different self-reports";

- elaborate on the vague wording in this context, namely "impairments";

- be more specific about the results.

Introduction:

- it is also a matter of urgency to show the theoretical background behind the distinction between the two types of depression (and therefore to justify intergroup comparisons);

- elaborate on those theoretical aspects that are subsequently studied (e.g., somatic anxiety or critical events) - there is neither sufficient reference to the literature nor a review of research in this area; 

- if you are writing (98-100) about the role of conscientiousness, please offer a specific thought on how it may influence the entry into motherhood; 

- similarly, verses 103-104 need to be developed in terms of an explanation of "how it works." 

Methods:

- the introduction of the 'SOS MAMMA' project is essential, but there is a need to show the importance of embedding the research  in this project;

- it is needed to show the importance of these details for the generalisability of the results;

- other methods besides questionnaires (e.g., metrics) are not described –   when reading about the group characteristics, we learn about different, but somehow collected, variables (e.g., the topic of alcohol); it would therefore be useful to add how information on these additional variables was obtained.

Results:

- in Table 2, the notation in the line 'Depressive symptoms (BDI-PRE-group / EPDS-POST-group) suggests as if one group was tested with a different tool than the other on depression and the results were compared; if this is true, it is a serious error; if it is a clarification problem, please pay attention to the overall explanation (also elsewhere in the text) of what the study did and how to understand the results;

- I suggest that the description of the results be enriched somewhat by clearly showing which factors turned out to be important in explaining levels of depression. 

Discussion

- add a 'Limitations' section and review in depth which factors in the design of the survey, selection of participants, data collection, and analysis of the data limit the significance of the results

- rewrite the entire discussion - in its current form, it lacks structure and order; moreover, it contains errors such as considering the conscientiousness variable as a significant negative predictor of depression in both study groups, while it is only significant in the second (POST) group;

- the discussion imprecisely addresses the results, I very much encourage you to rethink the results. 

Author Response

The article covers the scientifically and practically important topic of perinatal depression. I consider the question of the determinants of perinatal depression to be essential and in need of both research and scientific reflection. The text submitted, although based on important research, needs improvement both on the surface side (English language, punctuation, standardization of the recording of statistics, the decision as to whether to write "0.2" or ".2", Cronbach's a vs. alpha, etc.), and on the structure of the scientific statement and the very issue of depression shown here. I recommend proof-reading, and I elaborate on the latter two issues in this review: 

We are grateful to the Reviewer 2 for appreciating our work. We also modified and corrected the manuscript following Reviewer 2 advices. We corrected the English language, punctuation, and other typos in a consistent way.

Abstract:

- provide a rationale for why the two groups will be compared (why they should differ in the factors determining the onset of depression);

- expand on the vague phrase "different self-reports";

- elaborate on the vague wording in this context, namely "impairments";

- be more specific about the results.

We appreciated reviewer’ suggestion and we change the abstract following all the above suggestions.

Introduction:

- it is also a matter of urgency to show the theoretical background behind the distinction between the two types of depression (and therefore to justify intergroup comparisons);

We thank the reviewer for giving us the opportunity to improve the manuscript with this necessary background.

- elaborate on those theoretical aspects that are subsequently studied (e.g., somatic anxiety or critical events) - there is neither sufficient reference to the literature nor a review of research in this area; 

We thank the reviewer; in this revised version we insert background about anxiety and stressful events dimensions.

- if you are writing (98-100) about the role of conscientiousness, please offer a specific thought on how it may influence the entry into motherhood; 

We appreciated reviewer’ suggestion and we improve this part regarding the conscientiousness.

- similarly, verses 103-104 need to be developed in terms of an explanation of "how it works." 

Following the Reviewer suggestion, we detailed why the study of personality traits may influence the clinical intervention for perinatal depression.

Methods:

- the introduction of the 'SOS MAMMA' project is essential, but there is a need to show the importance of embedding the research  in this project;

We agree with the reviewer and we are sorry for the redundancy that we deleted.

- it is needed to show the importance of these details for the generalisability of the results;

Fo9llowing the Reviewer’ suggestion, we were more concise about the role of SOS MAMMA project.

- other methods besides questionnaires (e.g., metrics) are not described –   when reading about the group characteristics, we learn about different, but somehow collected, variables (e.g., the topic of alcohol); it would therefore be useful to add how information on these additional variables was obtained.

We are sorry and we insert the description of how sociodemographic and anamnestic details were obtained

Results:

- in Table 2, the notation in the line 'Depressive symptoms (BDI-PRE-group / EPDS-POST-group) suggests as if one group was tested with a different tool than the other on depression and the results were compared; if this is true, it is a serious error; if it is a clarification problem, please pay attention to the overall explanation (also elsewhere in the text) of what the study did and how to understand the results;

Two different tools were used. Both measure depressive symptoms, the BDI is appropriate for measuring depressive symptoms in adults regardless of life status, and the EPDS is appropriate and validated for measuring postpartum depressive symptoms and has not been validated for measuring depressive symptoms during the pregnancy. Therefore we consider it valid in terms of construction and coherence of the PRE and POST partum evaluation times to use two different tools.

- I suggest that the description of the results be enriched somewhat by clearly showing which factors turned out to be important in explaining levels of depression. 

We appreciate your suggestion and we summarized and clearly stated our findings at the beginning of the discussion.

Discussion

- add a 'Limitations' section and review in depth which factors in the design of the survey, selection of participants, data collection, and analysis of the data limit the significance of the results OK

- rewrite the entire discussion - in its current form, it lacks structure and order; moreover, it contains errors such as considering the conscientiousness variable as a significant negative predictor of depression in both study groups, while it is only significant in the second (POST) group;

- the discussion imprecisely addresses the results, I very much encourage you to rethink the results. 

We thank the Reviewer who gave us the opportunity to re-write the discussion and to insert the limitation section. We confirmed that the conscientiousness was a predictor of depression in both group and, following the reviewer suggestion, we improve the discussion about this point.